# Comparison of Regulatory T-Cell Subpopulations in Antithymocytic Globulin Versus Post-Transplant Cyclophosphamide for Preventing Graft-Versus-Host Disease in Allogeneic Hematopoietic Stem Cell Transplantation—A Retrospective Study

**DOI:** 10.3390/ijms26062521

**Published:** 2025-03-11

**Authors:** Bu-Yeon Heo, Jeong Suk Koh, Su-Young Choi, Thi Thuy Duong Pham, Sang-Woo Lee, Jung-Hyun Park, Yunseon Jang, Myung-Won Lee, Seul-Bi Lee, Wonhyoung Seo, Deog-Yeon Jo, Jaeyul Kwon, Ik-Chan Song

**Affiliations:** 1Department of Medical Science, College of Medicine, Chungnam National University, Daejeon 35015, Republic of Korea; xeyk1603@naver.com (B.-Y.H.); jdd02287@naver.com (S.-Y.C.); phamduong290194@gmail.com (T.T.D.P.); lleesy98@naver.com (S.-W.L.); 2Brain Korea 21 FOUR Project for Medical Science, College of Medicine, Chungnam National University, Daejeon 35015, Republic of Korea; 3Department of Internal Medicine, College of Medicine, Chungnam National University, Daejeon 34134, Republic of Korea; goldjs2323@naver.com (J.S.K.); iyoo23@naver.com (M.-W.L.); drizzle98@naver.com (S.-B.L.); bluedays13@naver.com (W.S.); deogyeon@cnu.ac.kr (D.-Y.J.); 4Translational Immunology Institute, College of Medicine, Chungnam National University, Daejeon 35015, Republic of Korea; jhpark76@cnu.ac.kr (J.-H.P.); harhie@naver.com (Y.J.)

**Keywords:** regulatory T cell, post-transplant cyclophosphamide, antithymocyte globulin, GVHD

## Abstract

Antithymocytic globulin (ATG) and post-transplant cyclophosphamide (PTCy) are frequently used regimens for graft-versus-host disease (GVHD) prophylaxis. However, there is a lack of data about the difference in regulatory T-cell (Treg) subpopulations between these two regimens. Peripheral blood samples were collected on day +21 following allogeneic hematopoietic stem cell transplantation (Allo-HSCT), and the Treg subpopulations were analyzed using flow cytometry. The Treg populations were categorized into three distinct subgroups: naïve, effector, and non-suppressive. And we compared overall survival (OS), the cumulative incidence of acute and chronic GVHD, and the relapse rate between the ATG and PTCy groups. We enrolled 45 patients (28 in ATG, 17 in PTCy) in total. In the ATG group, 16 and 12 patients underwent human leukocyte antigen (HLA) matched-sibling donor and unrelated donor HSCT, respectively. In the PTCy group, 12 patients underwent haplo-identical HSCT, and 5 patients underwent HLA-matched unrelated donor HSCT. The cumulative incidence of Grade 2–4 acute GVHD was 18.3% in the ATG group compared to 38.1% in the PTCy group (*p* = 0.13), while severe chronic GVHD occurred in 19.4% of ATG patients and 41.7% of PTCy patients (*p* = 0.343). And OS and the relapse rate were not statistically different between the two groups. The conventional CD25^+^FOXP3^+^Treg count of CD4 + T cells was higher in the PTCy group than in the ATG group (*p* = 0.0020). The effector Treg subset was significantly higher in the PTCy group than in the ATG group (*p* = 0.0412). And the effector Treg cell count had an inverse correlation with the severity of acute GVHD (*p* = 0.0007). Effector Tregs may be used as a biomarker to predict the severity of acute GVHD after allo-HSCT.

## 1. Introduction

Allogeneic hematopoietic stem cell transplantation (HSCT) is a valuable curative therapy for hematological malignancies. However, graft-versus-host disease (GVHD) remains of concern and is the leading cause of non-relapse mortality [1,2]. Antithymocytic globulin (ATG) and post-transplant cyclophosphamide (PTCy) were the most popularly used regimens for the prevention of acute and chronic GVHD [3,4]. ATG is a polyclonal T-cell-depleting antibody that targets T-cell antigens. It has been shown to significantly improve overall survival and GVHD-free survival compared to patients who did not receive ATG, particularly in those undergoing HLA-matched donor HSCT [5,6]. PTCy also effectively eradicates alloreactive T cells after infusion of haplo-identical stem cells. However, PTCy can preserve hematopoietic stem cells, reducing the incidences of acute and chronic GVHD [7,8].

ATG and PTCy had comparable clinical outcomes in terms of overall survival, relapse rate, and the incidence of acute or chronic GVHD in the setting of HLA-matched or haplo-identical HSCT [9,10,11]. However, few studies have been reported on their biological features, especially regulatory T (Treg) cells between ATG and PTCy. Treg cells are a functionally distinct subset of mature T cells with broad suppressive activity and which play a vital role in maintaining immune tolerance and suppressing potentially harmful immune responses after allogeneic HSCT [12,13]. Treg cells are the most important component in the development of GVHD and are considered therapeutic targets for various agents such as interleukin-2 (IL-2) [14,15,16]. It has been demonstrated that infusion of bone marrow cells without CD4^+^CD25^hi^Treg cells from transgenic donor mice during bone marrow transplantation accelerates the development of GVHD and decreases survival in an experimental animal model [17]. On the other hand, the transplantation of CD4^+^CD25^hi^Treg cells in a mouse model of allogeneic bone marrow transplantation was observed to abrogate GVHD [18]. And Kennedy-Nasser et al. demonstrated that ultra-low-dose IL-2 expands a Treg population in vivo and may be associated with a lower incidence of GVHD [15]. And the adoptive transfer of ex vivo expanded Treg cells had clinical improvement with substantially decreased steroid-refractory GVHD activity [19].

Treg cells are differentiated from CD4 + T cells and classically identified by the expression of the FOXP3 or IL-2 receptor α chain (CD25). As a way to consider the heterogeneity of the Treg compartment and analyze the properties of the Treg subpopulation, CD45RA was introduced to discriminate between antigen-experienced Treg (e.g., CD45RA^−^) and naïve Treg (e.g., CD45RA^+^) cells [20,21]. Treg subsets have been extensively studied to analyze the pathophysiology of autoimmune diseases such as systemic lupus erythematosus, rheumatoid arthritis (RA), and Behçet’s disease [22,23,24]. For example, in a study conducted on patients with RA, there was no difference in the total Treg population in the peripheral blood of RA patients and healthy controls, but the effector Treg, defined as CD45RA^−^CD25^hi^ population, was significantly decreased in RA patients compared to healthy individuals [25]. There is a lack of data analyzing Treg cell subpopulations between ATG and PTCy group for the GVHD prophylaxis. Therefore, this study aimed to compare the subset of Treg cells between these two regimens in patients with hematologic malignancies undergoing allogeneic HSCT.

## 2. Results

### 2.1. Patient Characteristics

The demographic characteristics of the patients are given in Table 1. Between June 2018 and June 2023, 45 patients were included in the study. In the PTCy group, 17 patients were enrolled, 5 with HLA-matched donors and 12 with HLA haplo-identical donors, while in the ATG group, all 28 patients were enrolled with HLA-matched donors. The median age for all patients was 54 years, with a range of 21 to 71 years. There were no differences between both groups in terms of gender, median age, the types and statuses of diseases, hematopoietic stem cell transplantation comorbidity index (HCT-CI), type of conditioning regimes, median stem cell dose infused, and median follow-up duration. However, CMV reactivation was significantly higher in the PTCy group compared to the ATG group (76.5% vs. 28.6%, *p* = 0.002).

### 2.2. GVHD

The cumulative incidence (CI) of grade II-IV acute GVHD at 100 days post transplantation was 18% (95% CI: 12–20%) in the ATG group, and 38% (95% CI: 10–42%) in the PTCy group (Figure 1A). The CI of chronic GVHD at 1 year was 67% in the ATG group and 71% in the PTCy group (*p* = 0.872; Figure 1B). There were similar results regarding the CI of severe degree chronic GVHD between both groups (19% vs. 42%, *p* = 0.343, Figure 1C).

### 2.3. Survival Outcomes

With the median follow-up of 19.0 months (range, 1.8–53.8 months) for all patients, there was no significant difference in the CI of relapse at 1 year between the two groups (ATG vs. PTCy, 22.6% vs. 35.4%; *p* = 0.106; Figure 2A). The 1-year CI of NRM was 11.7% for the ATG group and 13.1% for the PTCy group (*p* = 0.274; Figure 2B). Infections and GVHD were the most common causes of NRM, with no significant difference in the causes of death between the two GVHD prophylaxis groups. The 1-year OS was 77.4% for the ATG group and 62.1% for the PTCy group (*p* = 0.266; Figure 2C). Similarly, there was no significant difference in the 1-year GFRS between the two groups (64.1% vs. 44.8%, respectively, *p* = 0.401; Figure 2D).

### 2.4. Regulatory T Cells and Their Subpopulations

First, we analyzed the conventional Treg cells, defined as CD4^+^CD25^+^FOXP3^+^ in both the ATG and PTCy groups (Figure 3A). The proportion of conventional Treg cells was statistically significantly higher than in the PTCy group compared to the ATG group (12.96% in PTCy vs. 6.14% in ATG group, *p* = 0.0020; Figure 3B). Similar results were observed when defining Treg cells as CD25^+^CD127^low^ (7.44% in PTCy vs. 3.06% in the ATG group, *p* = 0.0007; Figure 3C). However, conventional Treg cells were not significantly associated with clinical outcomes such as CMV reactivation, relapse, acute GVHD and chronic GVHD (Figure 3D). When Treg cells were divided into three subpopulations based on CD45RA and FOXP3, the effector Treg subset was significantly higher in the PTCy group compared to the ATG group, while other subsets did not differ between the two groups. There were statistically significant differences between the two groups in the effector Treg subset (Figure 4A,B).

### 2.5. The Association Between Active Treg Cells and Clinical Outcomes

Next, we analyzed the correlation between each Treg subset and clinical outcomes. Naive Tregs (subset I), naïve CD4 + T cells (subset VI), non-suppressive Tregs (subset III), and subset IV did not correlate with clinical outcomes (Figure 5A,C,D,F). However, effector Treg cells (subset II) were significantly lower in patients with Grade II–IV acute GVHD compared to those with acute GVHD grade 0–I (*p* = 0.0154; Figure 5B). Conventional CD4 + T cells (Subset V) were statistically significantly increased in patients with grade II to IV acute GVHD compared to those with acute GVHD grade 0–I and mild to moderate chronic GVHD compared to those without chronic GVHD (Figure 5E). And, in the Spearman correlation analysis, an inverse correlation was observed between the severity of acute GVHD and the effector Treg cells (*p* = 0.007; Figure 6B). Furthermore, a positive correlation was observed between conventional CD4 + T cells (subset V) and the severity of acute GVHD (Figure 6E). However, other subsets did not correlate with the severity of acute GVHD (Figure 6A,C,D,F).

### 2.6. Inflammatory Cytokine Levels

The serum levels of inflammatory cytokines, including IL-6, IFN-γ, and TNF-α were measured to evaluate the immunological environment and its impact on Treg cells in the ATG and PTCy groups. The IL-6 levels were significantly elevated in the PTCy group compared to the ATG group (*p* = 0.0001; Figure 7A), indicating a stronger inflammatory milieu in the PTCy-treated patients. While the differences in IFN-γ and TNF-α levels did not reach statistical significance (*p* = 0.0782 and *p* = 0.0586, respectively), we observed a trend toward a higher inflammatory response in the PTCy group (Figure 7B,C).

## 3. Discussion

In this study, we found that using PTCy as a prophylactic regimen for GVHD preserves Treg cells better than using ATG. Among the patients included in this study, all patients of the ATG group received allogeneic HSCT from HLA-matched donors, while the PTCy group had approximately 70% of patients receiving haplo-identical donor HSCT. The fact that the PTCy group had a high number of haploidentical HSCT patients and haplo-identical HSCT is disadvantageous in terms of the occurrence of GVHD compared to HLA-matched donor HSCT, but there were no significant differences in clinical outcomes, suggesting that the use of PTCy is more effective in preventing GVHD by preserving Treg cells. In our study, serum IL-6 levels were significantly higher in the PTCy group than in the ATG group. This is because the PTCy group had many patients with haploidentical HSCT. Haploidentical HSCT is associated with more severe inflammation at the early period of transplantation and more frequent cytokine release syndrome than HLA-matched donor HSCT [26,27,28]. IL-6 is known not only to inhibit TGFβ-induced T-cell differentiation into regulatory T cells but also downregulate FOXP3 expression on Treg cells [29,30]. IL-6 is also considered a potential biomarker of acute GVHD after allogeneic HSCT, and elevated IL-6 levels have been shown to be significantly associated with worse outcomes, including severe cytokine release syndrome (CRS) and acute GVHD, and decreased overall survival [31,32]. However, in this study, despite higher IL-6 levels in the PTCy group, the incidence and severity of GVHD was similar to the ATG group, which may be due to better preservation of Tregs with PTCy, resulting in higher Treg levels in the PTCy group.

In the EBMT registry study for comparing ATG and PTCy in acute myeloid leukemia (AML) patients who underwent haplo-identical HSCT, patients in the PTCy group had significantly less grade III-IV acute GVHD than those in the ATG group [33]. And patients receiving PTCy had better GVHD-free, relapse-free survival and leukemia-free survival than those in the ATG group. In a meta-analysis study, PTCy also demonstrates a more favorable effect in preventing acute GVHD and improving overall survival (OS) compared to ATG [34]. This is thought to be attributed not only to the effective removal of alloreactive T cells by PTCy after allogeneic HSCT but also to the better recovery of Treg cells in the PTCy group, which was observed in our study. Rambaldi et al. also reported that the recovery of Treg cells after haplo-identical HSCT using PTCy was earlier than that after HLA-identical HSCT, resulting in a significantly higher Treg–conventional T-cell ratio during the early period after HSCT [35].

The results of this study alone are insufficient to fully elucidate the pathophysiology of why PTCy preserves Treg cells more effectively compared to ATG. However, by integrating findings from previously reported studies, it can be explained that rabbit ATG acts not only on alloreactive T cells but also on Treg cells, and its long half-life sustains this effect for an extended period. Consequently, due to the nature of ATG, Treg cells are presumed to undergo depletion following HSCT. On the other hand, PTCy primarily affects actively proliferating alloreactive T or NK cells and spares hematopoietic stem cells or Treg cells by their high expression of aldehyde dehydrogenase [36]. In the murine experiment conducted by Ganguly et al., the rapid recovery of donor-derived Tregs was observed after PTCy treatment, suggesting its potential role in GVHD prevention [37]. This suggests that PTCy-based GVHD prophylaxis preferentially promotes the reconstitution of Tregs in a clinical setting. In a previous study on post-transplant complications associated with Treg cells that we reported, it was observed that in most patients treated with PTCy after allo-HSCT, naïve, and effector Treg cells were well-preserved. However, in some patients with insufficient Treg cells, the manifestation of life-threatening GVHD or complications such as autoimmune limbic encephalitis was observed [38].

As effective as PTCy is in preventing GVHD, there are still challenges that need to be addressed. Firstly, despite the use of PTCy, some patients still experience severe GVHD or CRS [39]. To address this issue, recent approaches include combining PTCy and ATG, which has been reported to be more effective in reducing GVHD than either alone. [26,40,41]. Second, T-cell depletion appears to occur more frequently with PTCy compared to patients not using PTCy, which, along with high levels of Treg cells, contributes to a higher risk of relapse. For example, in the EBMT registry data, we observe a slightly higher recurrence rate in the PTCy group compared to the ATG group [33]. Therefore, for patients at high risk of relapse after allo-HSCT, additional efforts will be needed to prevent relapse, such as donor lymphocyte infusion (DLI) or maintenance therapy.

In this study, CMV reactivation was observed more frequently in the PTCy group than in the ATG group. This is similar to the results of previous reports. In a study of HLA-matched donor HSCT using ATG, Kröger et al. reported that CMV reactivation occurred in 21% of patients, and Choräo et al. reported that CMV reactivation was observed in 66% of haploidentical HSCT using PTCy [3,42]. This is thought to be because PTCy suppresses alloreactive T cells, which leads to more frequent CMV reactivation [7]. However, in this study, CMV reactivation was effectively controlled with preemptive therapy using ganciclovir, and there was no mortality due to CMV disease. This study has several limitations. First, the stem cell donors in the ATG group and the PTCy group are different. This is because this study is a retrospective analysis comparing two GVHD prophylaxis regimens used in real clinical practice. In general, ATG is used in HLA-matched donor HSCT, and PTCy is used in haplo-identical HSCT [3,6,43]. And, PTCy has recently been proven to be effective in preventing GVHD and is also being used in some cases of HLA-matched donor HSCT [44,45]. Second, this study analyzed Tregs at a single time point on the 21st day after allo-HSCT. In order to analyze the dynamics of GVHD and its impact on chronic GVHD, it will be necessary to analyze Tregs serially in the future. However, the same time point analysis in this study is thought to be more clinically useful in deriving predictive biomarkers.

To our knowledge, this study is the first report to compare subpopulations of Treg cells in patients undergoing allogeneic HSCT when using ATG and PTCy. Furthermore, given the inverse correlation observed between effector Treg cells measured in peripheral blood early post transplant and the severity of acute GVHD, this could be used as a biomarker to predict the severity of acute GVHD after allogeneic HSCT. Therefore, in patients at high risk of acute GVHD due to low levels of effector Treg cells, interventions such as increasing the dose of immunosuppressive agents or administering a GVHD treatment agent such as ruxolitinib earlier should be considered.

In conclusion, the use of PTCy preserves Treg cells more effectively compared to using ATG, and among the Treg subpopulations, effector Treg cells exhibit an inverse correlation with the severity of acute GVHD. Therefore, effector Tregs can be used as a biomarker to predict the severity of acute GVHD after allo-HSCT.

## 4. Materials and Methods

### 4.1. Patients and Treatments

We retrospectively analyzed consecutive adult patients (age, >18 years) with hematological malignancies who underwent allogeneic HSCT in Chungnam National University Hospital (Daejeon, Republic of Korea) between June 2018 and June 2023. We excluded those receiving second transplantations and patients with refractory disease. PTCy was given on days +3 and +4 at a dose of 50 mg/kg according to Johns Hopkins protocol [7]. Rabbit ATG (thymoglobulin; Sanofi-Aventis, Paris, France) was given from days −3 to −1 at a dose of 1.5 mg/kg based on a previous pivotal study [46]. We usually assign PTCy in haplo-identical HSCT and ATG in HLA-matched donor HSCT for prophylaxis of GVHD according to the institute’s policy. Two conditioning regimens were used. In the myeloablating conditioning (MAC) regimen, 3.2 mg/kg busulfan was administered for 4 days and 40 mg/m^2^ fludarabine was administered for 5 days. In the reduced intensity conditioning (RIC) regimen, 3.2 mg/kg busulfan was administered for 2 days and 30 mg/m^2^ fludarabine was administered for 6 days. RIC was administered to patients over 55 years of age or with comorbidities. No pharmacokinetic adjustment of busulfan dose was performed. Cyclosporine or tacrolimus for GVHD prophylaxis was given commencing on day −1 in the ATG group and on day +5 in the PTCy group. All patients received granulocyte colony-stimulating factor-mobilized peripheral blood stem cells (PBSCs; target CD34+ cell count, 5 × 10^6^/kg). Filgrastim 300 ug/m^2^ was administered from day +5 until neutrophil recovery. No therapy for the prevention of relapse after allogeneic HSCT, such as donor lymphocyte infusion or hypomethylating agents or tyrosine kinase inhibitors, was added.

### 4.2. Clinical Outcomes

We collected clinical data for assessing the overall survival (OS), the incidences and severity of acute and chronic GVHD, the relapse rate, non-relapse mortality (NRM), and cytomegalovirus (CMV) and Epstein–Barr virus (EBV) reactivation. Acute GVHD was graded using the Mount Sinai Acute GVHD International Consortium (MAGIC) criteria, and chronic GVHD was graded according to the National Institutes of Health (NIH) consensus [47,48]. And GVHD-free, relapse-free survival (GFRS) was defined as the occurrence of any of the following events from the time of transplantation: grade III or IV acute GVHD, chronic GVHD warranting systemic immunosuppression, disease relapse or progression, or death from any cause. NRM was defined as death from any cause other than relapse. CMV and EBV reactivation was defined as the detection of viral DNA in whole blood by PCR at least once.

### 4.3. Assessment of Regulatory T Cells Subpopulation and Cytokine

Peripheral blood mononuclear cells (PBMCs) were obtained from whole blood at day +21 after allogeneic HSCT using lymphocyte separation medium (Corning, New York, NY, USA) by density gradient centrifugation. The PBMCs were stained with live/dead fixable stain dye (Life technologies, Carlsbad, CA, USA) to distinguish live and dead cells. After PBS washing, the cells were incubated with FITC-CD3 (BD Biosciences, Franklin Lakes, NJ, USA), PerCP-Cy5.5-CD4 (BD Biosciences, Franklin Lakes, NJ, USA), BV421-CD25 (BD Biosciences, Franklin Lakes, NJ, USA), APC-CD127 (Biolegend, San Diego, CA, USA), and PE-Cy7-CD45RA (BD Biosciences, Franklin Lakes, NJ, USA). The cells were then fixed and permeabilized with Foxp3/Transcription Factor Staining Buffer Set (eBioscience, San Diego, CA, USA) and further stained with PE-Foxp3 (BD Biosciences, Franklin Lakes, NJ, USA). Based on markers CD25 and CD45RA, the subpopulation of Treg are as follows: CD25^int^CD45RA^+^ cells (Subgroup I, naive/resting Treg).

CD25^hi^CD45RA^−^ cells (Subgroup II, activated/effector Treg), CD25^int^CD45RA^−^ cells (Subgroup III, non-suppressive Treg), CD25^low^CD45RA^−^ cells (Subgroup IV), CD25^−^CD45RA^−^cells (Subgroup V, effector Tconv), and CD25^−^CD45RA^+^ cells (Subgroup VI, naïve Tconv) (Figure 1). Treg cells and their subpopulation were analyzed with a FACSCanto II flow cytometer (BD Biociences), and data were processed with FlowJo software v10.10 (Tree Star, OR, USA). Interleukin-6 (IL-6), tumor necrosis factor-alpha (TNF-α), and interferon-gamma (IFN-γ) were analyzed by enzyme-linked immunosorbent assay of plasma samples.

### 4.4. Statistical Analysis

Categorical variables were compared using the chi-squared test and logistic regression was employed to examine correlations. Overall and leukemia-free survival was assessed using the Kaplan–Meier method. Survival rates were compared using the log-rank test. Cumulative incidence functions were used to estimate the acute and chronic GVHD rates, relapse rate, and NRM. A *p*-value < 0.05 was considered to reflect significance. All statistical analyses were performed with the aid of SPSS software version 24.0 (IBM Corporation, Armonk, NY, USA).

### 4.5. Ethics Statement

The study protocol was approved by the Institutional Review Board of Chungnam National University Hospital (IRB No. CNUH 2018-08-013-012). Written informed consent was obtained from all patients.

## Figures and Tables

**Figure 1 ijms-26-02521-f001:**
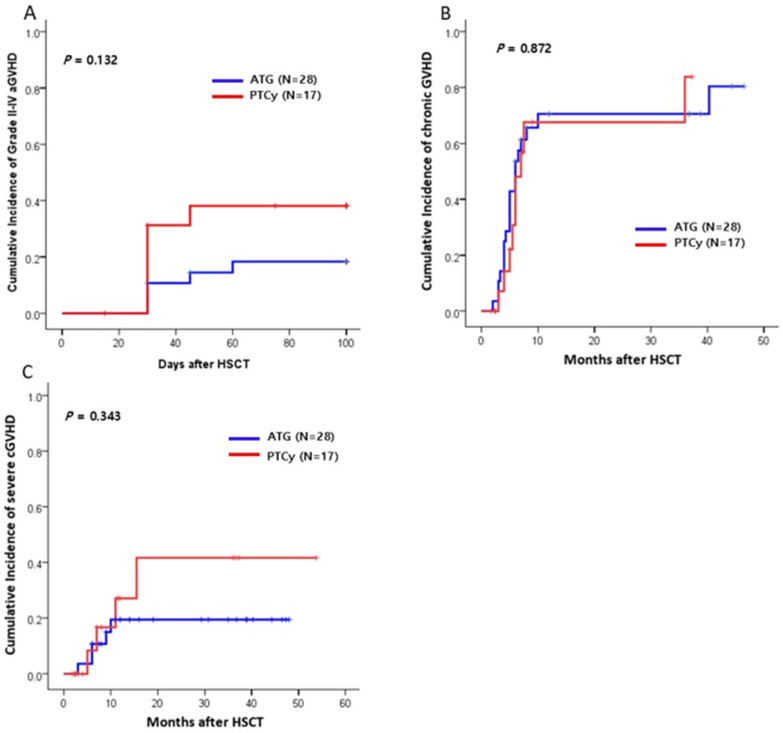
Cumulative incidences (CIs) of graft-versus-host disease (GVHD) between ATG and PTCy group (n = 45). (**A**) The CI of grade II–IV acute GVHD (aGVHD). (**B**) The CI of chronic GVHD (cGVHD). (**C**) The CI of severe chronic GVHD. HSCT, hematopoietic stem cell transplantation.

**Figure 2 ijms-26-02521-f002:**
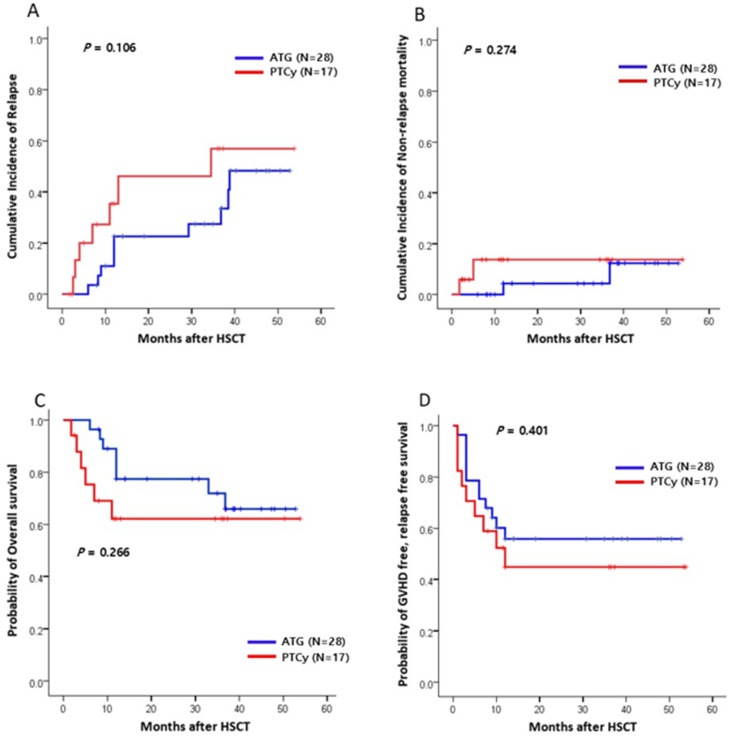
Clinical outcomes between the ATG and PTCy groups (n = 45). (**A**) The CI of relapse (OS). (**B**) The CI of non-relapse mortality (NRM). (**C**) The probability of overall survival (OS). (**D**) The probability of GVHD-free, relapse-free survival (GRFS). HSCT, hematopoietic stem cell transplantation.

**Figure 3 ijms-26-02521-f003:**
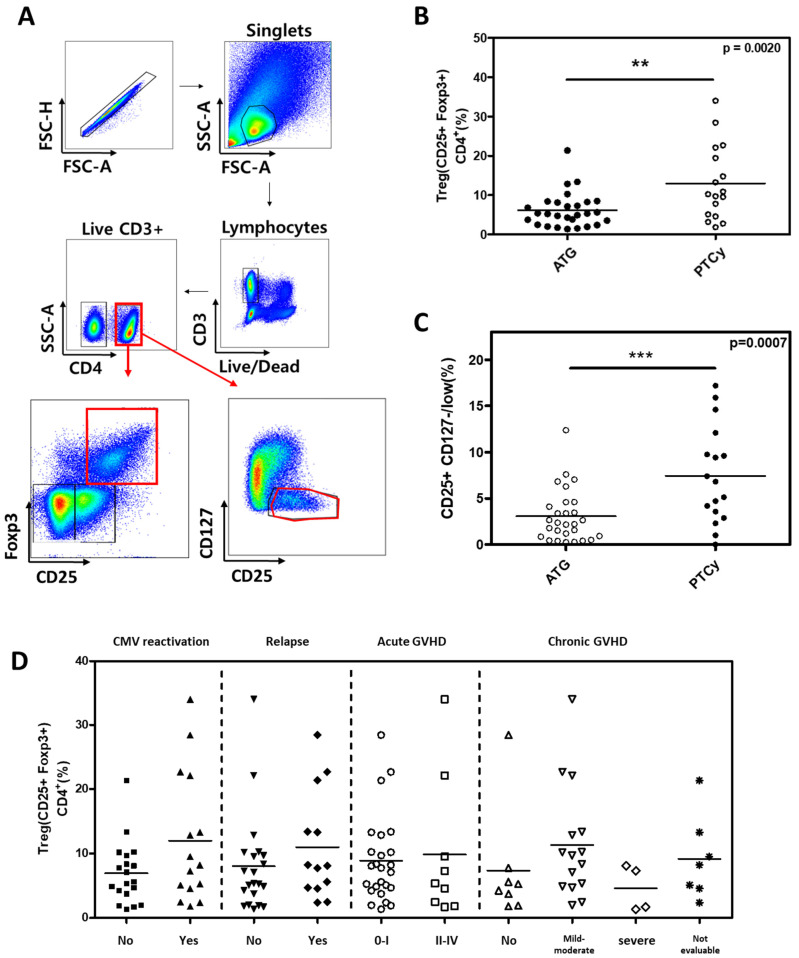
Comparison of Treg cells between ATG and PTCy groups and association between Treg cells and clinical outcomes. (**A**) Flow cytometry gating strategy, with CD4 + T cells divided into CD25^+^Foxp3^+^ and CD25^+^ CD127^−/low^ (**B**) Proportions of CD25^+^Foxp3^+^ Treg cells among CD4 + T cells were compared between ATG and PTCy patients. (**C**) Proportions of CD25^+^ CD127^−/low^ Treg cells among CD4 + T cells were compared between ATG and PTCy patients. (**D**) Association between CD25^+^Foxp3^+^ Treg cells and clinical outcomes, such as CMV reactivation, relapse, acute GVHD, and chronic GVHD. Statistical differentiation by two-tailed *t*-test. ** *p* < 0.01, *** *p* < 0.001.

**Figure 4 ijms-26-02521-f004:**
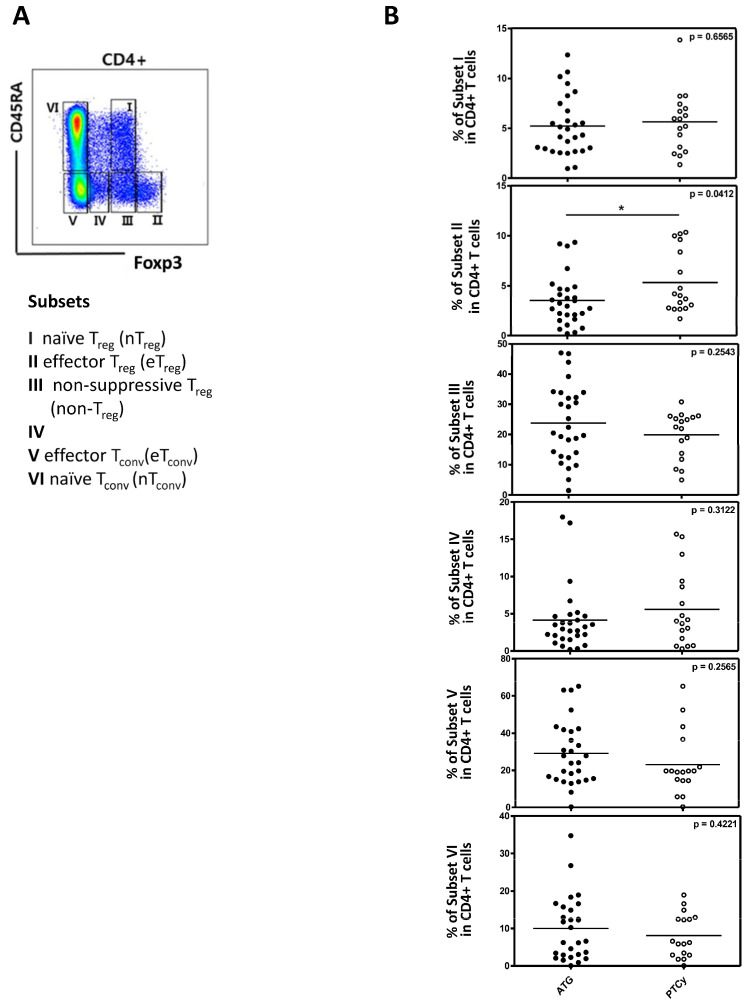
The analysis of Treg cell subpopulations between the ATG and PTCy groups (n = 45). (**A**) A flow cytometry gating strategy with division into subsets I-VI based on CD45RA and FOXP3 as indicated. (**B**) A comparison of CD4 + T cells between the ATG and PTCy groups in each subset. Statistical differentiation by two-tailed *t*-test. * *p* < 0.05.

**Figure 5 ijms-26-02521-f005:**
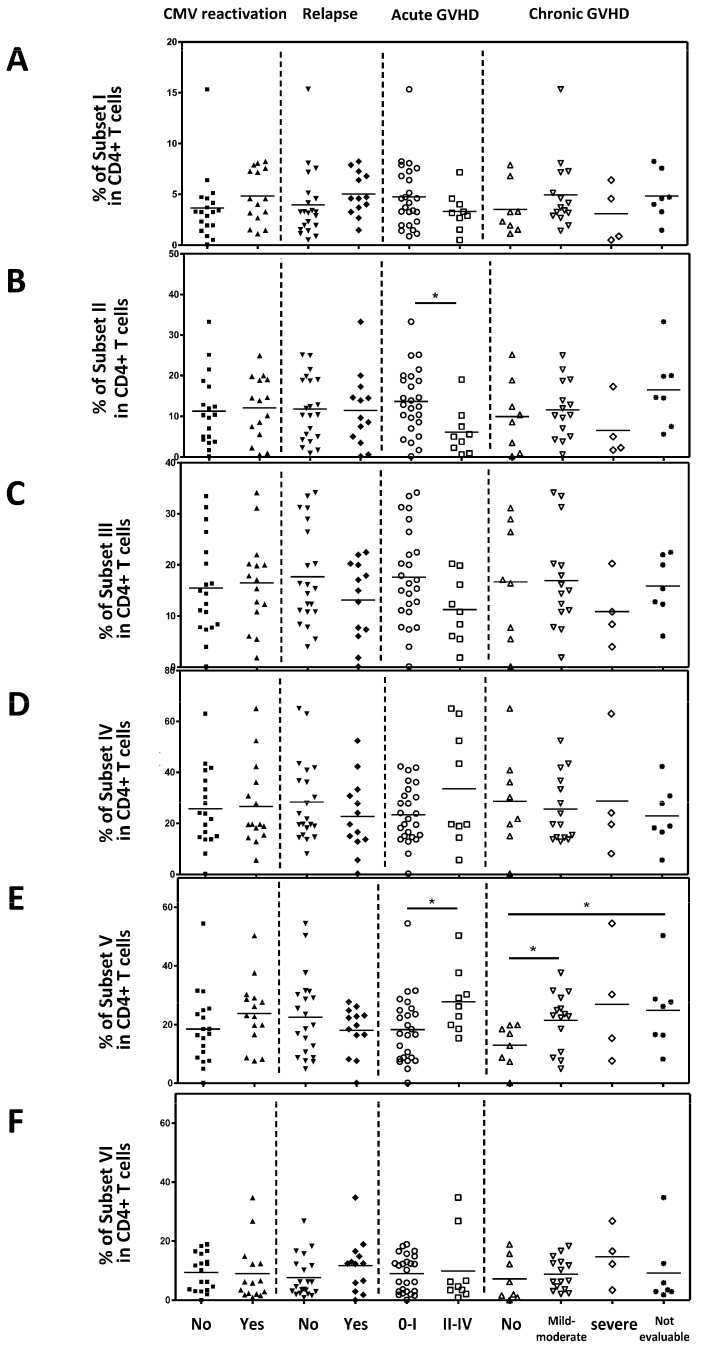
The association between the Treg cell subset and the clinical outcomes. (**A**) Naïve Treg cells. (**B**) Effector Treg cells. (**C**) Non-suppressive Treg cells. (**D**) Subset IV. (**E**) Conventional CD4 + T cells. (**F**) Naïve CD4 + T cells. Statistical differentiation by two-tailed *t*-test. * *p* < 0.05.

**Figure 6 ijms-26-02521-f006:**
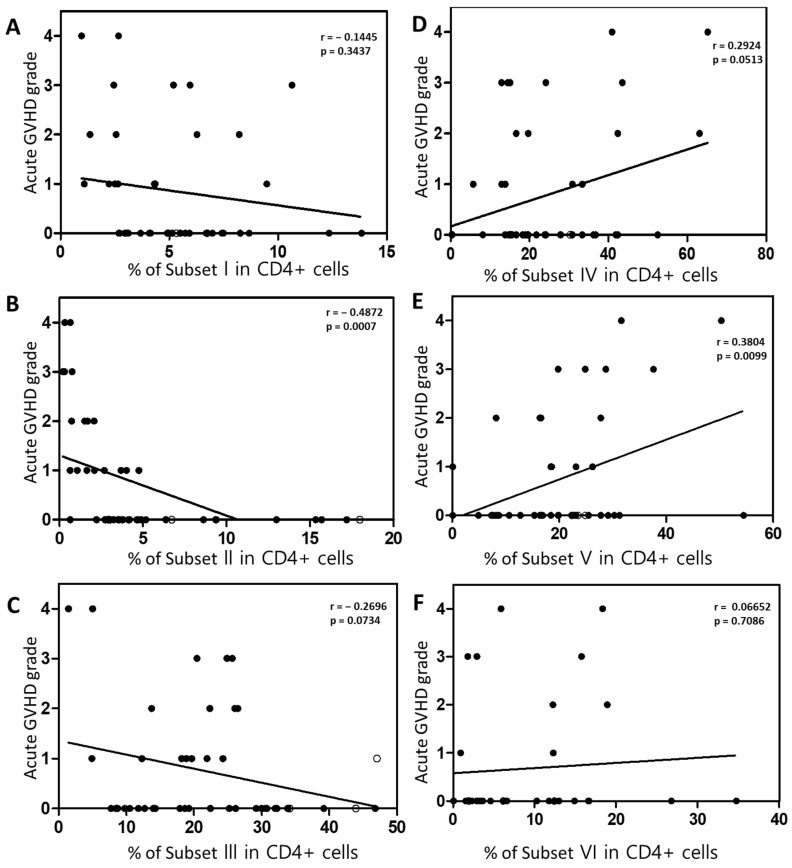
Spearman correlation analysis between each subset and the severity of acute GVHD. (**A**) Naïve Treg cells. (**B**) Effector Treg cells. (**C**) Non-suppressive Treg cells. (**D**) Subset IV. (**E**) Conventional CD4 + T cells. (**F**) Naïve CD4 + T cells. Spearman’s correlation coefficient (r) and the *p* value are indicated.

**Figure 7 ijms-26-02521-f007:**
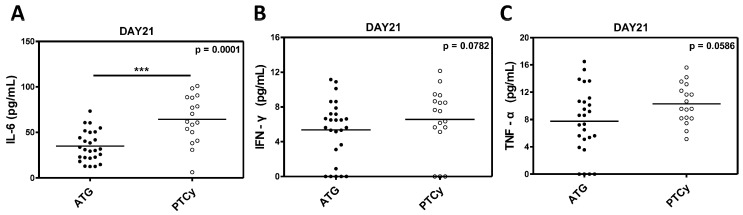
Inflammatory cytokine levels between ATG and PTCy groups. (**A**) IL-6. (**B**) IFN-γ. (**C**) TNF-α. Statistical difference by two-tailed *t*-test. *** *p* < 0.001.

**Table 1 ijms-26-02521-t001:** Clinical characteristics in patients with allogeneic-HSCT between ATG and PTCy group (n = 45).

	ATG (n = 28)	PTCy (n = 17)	*p*-Value
Median Age, year (range)	52.5 (21–66)	57 (29–71)	0.060
Gender, M/F	16/12	11/6	1.000
Type of diseases			0.824
Acute myeloid leukemia	18 (64.3%)	10 (58.8%)	
Acute lymphoblastic leukemia	6 (21.4%)	3 (17.6%)	
Myelodysplastic syndrome	4 (14.3%)	4 (23.5%)	
Type of donors			<0.001
HLA-matched sibling	16 (57.1%)	0 (0.0%)	
HLA-matched unrelated	12 (42.9%)	5 (29.4%)	
Haplo-identical	0 (0.0%)	12 (70.6%)	
Disease status at transplant			0.830
1st CR	18 (64.3%)	11 (64.7%)	
2nd CR	4 (14.3%)	1 (5.9%)	
MDS	4 (14.3%)	4 (23.5%)	
Persistent	2 (7.1%)	1 (5.9%)	
Poor risk *	15 (53.6%)	7 (43.8%)	0.755
HCT-CI			0.434
0	20 (71.5%)	13 (76.5%)	
1–2	8 (28.6%)	3 (17.6%)	
3–	0 (0.0%)	1 (5.9%)	
CMV reactivation	8 (28.6%)	13 (76.5%)	0.002
Acute GVHD (evaluable)			0.215
Grade 0–I	23 (82.1%)	11 (37.6%)	
Grade II–IV	5 (17.9%)	6 (35.3%)	
Stem cell source			-
PB	28 (100%)	17 (100%)	
BM	0 (0.0%)	0 (0.0%)	
Conditioning regimen			0.101
MAC	22 (78.6%)	9 (52.9%)	
RIC	6 (21.4%)	8 (47.1%)	
Cell count, median (range)			
TNC count (×10^8^ cells/kg)	11.97 (6.86–22.51)	12.08 (6.87–20.60)	0.926
CD34 + cell (×10^6^ cells/kg)	7.94 (2.60–22.17)	11.11 (2.17–36.00)	0.216
Median F/U duration, month (range)	16.8 (3.8–23.3)	11.5 (1.8–53.8)	0.101

* Poor risk includes sAML, tAML, AML with poor risk group in NCCN guidelines, poor cytogenetics in ALL. ATG, antithymocyte globulin; PTCy, posttransplant cyclophosphamide; HSCT, hematopoietic stem cell transplantation; CR, complete remission; MDS, myelodysplastic syndrome; HCT-CI, hematopoietic stem cell transplantation comorbidity index; CMV, cytomegalovirus; PB, peripheral blood; BM, bone marrow; MAC, myeloablative conditioning; RIC, reduced-intensity conditioning; TNC, total nucleated cell.

## Data Availability

Original data can be requested from the corresponding author (petrosong@cnu.ac.kr).

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
