# Peer review of "Comparison of Regulatory T-Cell Subpopulations in Antithymocytic Globulin Versus Post-Transplant Cyclophosphamide for Preventing Graft-Versus-Host Disease in Allogeneic Hematopoietic Stem Cell Transplantation—A Retrospective Study"

_ijms, 2025, doi:10.3390/ijms26062521_

Round 1

Reviewer 1 Report

Comments and Suggestions for Authors

In the submitted manuscript, the authors compred Treg subpopulations between antithymocytic globulin (ATG) and post-transplant cyclophosphamide (PTCy) regimens for GVHD prophylaxis in allogeneic HSCT. The authors collected peripheral blood from 45 patients (28 ATG, 17 PTCy) at 21 days post-transplantation and analyzed the abundance of Treg and and their correlation with clinical outcomes.

Overall, the study is very superficial and lacks clinical significance. There is a fundamental caveat within the study that prevents the interpretation of the results.

1. What is the rationale of analyzing Treg in HSCT? Why is it more important than other T cell subsets in HSCT? The authors did not provide sufficient background of the work. The motivation of the study if unclear. The clinical relevance of the observed differences in Treg subpopulations needs further elaboration.

2. There is a fundamental caveat in the study design. The ATG group is dominated by HLA-matched sibling donors, while the PTCy group is dominated by haploidentical HSCT. Since the study groups are not well matched, why do the authors assume that the results are not simply due to differences in the type of HSCT performed?

3. The study only analyzes a single time point on day 21. The follow-up period is short for assessing long-term outcomes.

Author Response

In the submitted manuscript, the authors compared Treg subpopulations between antithymocytic globulin (ATG) and post-transplant cyclophosphamide (PTCy) regimens for GVHD prophylaxis in allogeneic HSCT. The authors collected peripheral blood from 45 patients (28 ATG, 17 PTCy) at 21 days post-transplantation and analyzed the abundance of Treg and and their correlation with clinical outcomes.

Overall, the study is very superficial and lacks clinical significance. There is a fundamental caveat within the study that prevents the interpretation of the results.

  1. What is the rationale of analyzing Treg in HSCT? Why is it more important than other T cell subsets in HSCT? The authors did not provide sufficient background of the work. The motivation of the study if unclear. The clinical relevance of the observed differences in Treg subpopulations needs further elaboration.

Author’s answer) Thank you very much for your kind advice. This paper compares two regimens (ATG and PTCy) for the prevention of GVHD, a complication of allo-HSCT. The mechanism of GVHD is very complex and difficult to explain simply, but it is the process by which host tissues are damaged by immune cells derived from the donor. It is well known that regulatory T cells play the most important role in regulating the function of immune cells, and various studies have shown that Tregs are closely related to the occurrence of GVHD. Therefore, in this study, we subdivided Tregs and analyzed whether effector Tregs, a component of Tregs, are associated with the occurrence of GVHD. In accordance with the reviewer's comments, the content of the Introduction section has been supplemented and written as follows.

“Treg cells are a functionally distinct subset of mature T cells with broad suppressive activity and play a vital role in maintaining immune tolerance and suppressing potentially harmful immune responses after allogeneic HSCT. Treg cells are the most important component in the development of GVHD and are considered as therapeutic targets for various agents such as interleukin-2 (IL-2). It has been demonstrated that infusion of bone marrow cells without CD4+CD25hiTreg cells from transgenic donor mice during bone marrow transplantation accelerates the development of GVHD and decreases the survival in an experimental animal model. On the other hand, the transplantation of CD4+CD25hiTreg cells in a mouse model of allogeneic bone marrow transplantation was observed to abrogate GVHD.”

  1. There is a fundamental caveat in the study design. The ATG group is dominated by HLA-matched sibling donors, while the PTCy group is dominated by haploidentical HSCT. Since the study groups are not well matched, why do the authors assume that the results are not simply due to differences in the type of HSCT performed?

Author’s answer) ATG has been studied a lot in HLA-matched donor HSCT, and PTCy is a GVHD prophylaxis regimen originally developed for haplo-identical HSCT. Of course, recently, due to the excellent efficacy of PTCy, its use is increasing in HLA-matched donor HSCT as well. This study retrospectively analyzed two regimens used in real practice, so ATG was used only for HLA-matched donor HSCT, and PTCy was mostly used for haplo-identical HSCT. The difference in the baseline characteristics of patients in the two groups is due to the fact that this study retrospectively analyzed data from real practice. In accordance with the reviewer's comments, the content of the Discussion section has been supplemented and written as follows.

“This study has several limitations. First, the stem cell donors in the ATG group and the PTCy group are different. This is because this study is a retrospective analysis comparing two GVHD prophylaxis regimens used in real clinical practice. In general, ATG is used in HLA-matched donor HSCT, and PTCy is used in haplo-identical HSCT. And, PTCy has recently been proven to be effective in preventing GVHD and is also being used in some cases of HLA-matched donor HSCT.”

  1. The study only analyzes a single time point on day 21. The follow-up period is short for assessing long-term outcomes.

Author’s answer) The 21 days after HSCT is the period when the reconstitution of immune cells occurs. Various studies have attempted to examine samples of blood or bone marrow on the 21st day to evaluate the clinical outcomes of HSCT. This study also suggests the role of effector Tregs on the 21st day as a biomarker for predicting acute GVHD, and considering the clinical utility of biomarkers, it is recommended to perform the test at a fixed time.

The median follow up duration in this study was about 1 year, which we believe is sufficient time to measure the development of acute GVHD. This study found that Tregs at day 21 correlated with the development of acute GVHD. Although there are rare cases of new cases of chronic GVHD after 1 year, as shown in the figure in this study, there is no statistical difference between the two groups, so it is unlikely that there is a change in the tendency of chronic GVHD after 1 year. In addition, this study was conducted at a relatively early time point after allo-HSCT, so the main purpose of the study was to identify the correlation between Treg and acute GVHD. In accordance with the reviewer's comments, the content of the Discussion section has been supplemented and written as follows.

“Second, this study analyzed Tregs at a single time point on the 21st day after allo-HSCT. In order to analyze the dynamics of GVHD and its impact on chronic GVHD, it will be necessary to analyze Tregs serially in the future. However, the same-time point analysis in this study is thought to be more clinically useful in deriving predictive biomarkers.”

Reviewer 2 Report

Comments and Suggestions for Authors

Comments to the authors

The reviewer finds that the current title is somewhat lengthy. Could the title be shortened for clarity and conciseness? One possible suggestion is:

"Comparison of Regulatory T Cell Subpopulations in ATG vs. Post-Transplant Cyclophosphamide for Preventing GvHD in Allogeneic HSCT."

Lines 25 to 27: The current phrasing is somewhat fragmented. I recommend revising lines 25 to 27 as follows: "Peripheral blood samples were collected on day +21 following allogeneic hematopoietic stem cell transplantation (Allo-HSCT), and the Treg subpopulations were analyzed using flow cytometry. The Treg populations were categorized into three distinct subgroups: naïve, effector, and non-suppressive."

Lines 33 to 35: T The statistical results could be more clearly integrated into the results section, and it may also be helpful to specify the test used for comparison. I recommend revising as follows:

"The cumulative incidence of Grade 2-4 acute GVHD was 18.3% in the ATG group compared to 38.1% in the PTCy group (p = 0.13), while severe chronic GVHD occurred in 19.4% of ATG patients and 41.7% of PTCy patients (p = 0.343)."

Introduction:

In this section, some sentences are quite long, which makes the text more difficult to follow. Breaking them into shorter, more digestible parts could enhance readability. For example, lines 50 to 53. Maybe this sentence chould be revised as: ATG is a polyclonal T-cell depleting antibody that targets T-cell antigens. It has been shown to significantly improve overall survival and GVHD-free survival compared to patients who did not receive ATG, particularly in those undergoing HLA-matched donor HSCT.

2.1 Patient Characteristics

The significant difference in CMV reactivation rates between the PTCy and ATG groups is a notable finding. However, is there any additional context or explanation for why this might be the case? For example, could it be related to the type of conditioning regimen or the specific patient characteristics in each group? Is there any mention of treatment for CMV reactivation or any impact it may have had on GVHD or overall survival?

Lines 179 to 181: The significant elevation of IL-6 levels in the PTCy group suggests a stronger inflammatory response. Can you provide any further explanation as to why PTCy might induce such a strong inflammatory milieu compared to ATG? Is this consistent with previous studies, or are there any known mechanisms behind this increase in IL-6?

Lines 192 to 194: The statement mentions that the PTCy group received HSCT under “unfavorable conditions,” specifically with a higher proportion of haplo-identical donors. Can you clarify why this is considered unfavorable in the context of the study, and how it could potentially affect GVHD outcomes? Is there data from other studies that support this characterization of haplo-identical transplants being less favorable compared to HLA-matched transplants?

  1. Materials and Methods

Although I understand that the journal’s formatting requirements may dictate the current structure, I would recommend considering a more conventional format, where the Materials and Methods section is presented before the Results and Discussion. This sequence helps ensure that readers first understand the experimental design and methodology before interpreting the findings and drawing conclusions, enhancing the overall readability and logical flow of the manuscript.

Lines 259 to 261: Is there any rationale or literature supporting the choice of the specific doses (e.g., 50 mg/kg for PTCy, 1.5 mg/kg for ATG) used in this study? It could help to briefly explain whether these doses were based on institutional protocol or evidence from previous studies.

Author Response

The reviewer finds that the current title is somewhat lengthy. Could the title be shortened for clarity and conciseness? One possible suggestion is:

"Comparison of Regulatory T Cell Subpopulations in ATG vs. Post-Transplant Cyclophosphamide for Preventing GvHD in Allogeneic HSCT."

Author’s answer) We sincerely appreciate the reviewer's kind advice. We have changed the title as recommended by the reviewer.

Lines 25 to 27: The current phrasing is somewhat fragmented. I recommend revising lines 25 to 27 as follows: "Peripheral blood samples were collected on day +21 following allogeneic hematopoietic stem cell transplantation (Allo-HSCT), and the Treg subpopulations were analyzed using flow cytometry. The Treg populations were categorized into three distinct subgroups: naïve, effector, and non-suppressive."

Author’s answer) We have made changes to the abstract as recommended by the reviewer.

Lines 33 to 35: T The statistical results could be more clearly integrated into the results section, and it may also be helpful to specify the test used for comparison. I recommend revising as follows:

"The cumulative incidence of Grade 2-4 acute GVHD was 18.3% in the ATG group compared to 38.1% in the PTCy group (p = 0.13), while severe chronic GVHD occurred in 19.4% of ATG patients and 41.7% of PTCy patients (p = 0.343)."

Author’s answer) We have made changes to the abstract as recommended by the reviewer.

Introduction:

In this section, some sentences are quite long, which makes the text more difficult to follow. Breaking them into shorter, more digestible parts could enhance readability. For example, lines 50 to 53. Maybe this sentence chould be revised as: ATG is a polyclonal T-cell depleting antibody that targets T-cell antigens. It has been shown to significantly improve overall survival and GVHD-free survival compared to patients who did not receive ATG, particularly in those undergoing HLA-matched donor HSCT.

Author’s answer) We have made changes to the introduction section as recommended by the reviewer.

2.1 Patient Characteristics

The significant difference in CMV reactivation rates between the PTCy and ATG groups is a notable finding. However, is there any additional context or explanation for why this might be the case? For example, could it be related to the type of conditioning regimen or the specific patient characteristics in each group? Is there any mention of treatment for CMV reactivation or any impact it may have had on GVHD or overall survival?

Author’s answer) I am deeply grateful for the reviewer's careful advice. As you mentioned, the results of this study showed that the incidence of CMV reactivation was higher in the group that used PTCy. This is thought to be because PTCy eliminated alloreactive T cells, which caused CMV reactivation. The results of this study are similar to those of other previous studies, and there was no mortality due to CMV disease. The following has been added to the Discussion section to address your comments. In accordance with the reviewer's comments, the content of the Discussion section has been supplemented and written as follows.

“In this study, CMV reactivation was observed more frequently in the PTCy group than in the ATG group. This is similar to the results of previous reports. In a study of HLA-matched donor HSCT using ATG, Kröger et al. reported that CMV reactivation occurred in 21% of patients, and Choräo et al. reported that CMV reactivation was observed in 66% of haploidentical HSCT using PTCy. This is thought to be because PTCy suppresses alloreactive T cells, which leads to more frequent CMV reactivation. However, in this study, CMV reactivation was effectively controlled with preemptive therapy using ganciclovir, and there was no mortality due to CMV disease.”

Lines 179 to 181: The significant elevation of IL-6 levels in the PTCy group suggests a stronger inflammatory response. Can you provide any further explanation as to why PTCy might induce such a strong inflammatory milieu compared to ATG? Is this consistent with previous studies, or are there any known mechanisms behind this increase in IL-6?

Author’s answer) The high IL-6 levels in the PTCy group are thought to be due to the large number of haploidentical HSCT patients in this group, rather than being due to the effects of PTCy itself. Compared to HLA-matched donor HSCT, haploidentical HSCT is more likely to cause inflammation in the early stages of transplantation, and Cytokine release syndrome frequently occurs in this high inflammatory status. Among various cytokines, IL-6 is currently widely used in clinical practice as a predictive marker. In accordance with the reviewer's comments, the content of the Discussion section has been supplemented and written as follows.

“This is because the PTCy group had many patients with haploidentical HSCT. Haploidentical HSCT is associated with more severe inflammation at the early period of transplantation and more frequent cytokine release syndrome than HLA-matched donor HSCT”

Lines 192 to 194: The statement mentions that the PTCy group received HSCT under “unfavorable conditions,” specifically with a higher proportion of haplo-identical donors. Can you clarify why this is considered unfavorable in the context of the study, and how it could potentially affect GVHD outcomes? Is there data from other studies that support this characterization of haplo-identical transplants being less favorable compared to HLA-matched transplants?

Author’s answer) Haploidentical HSCT is disadvantageous in terms of the occurrence of GVHD compared to HLA-matched donor HSCT. Of course, recently, as methods for preventing GVHD have advanced (as in this study, using PTCy, the incidence of GVHD is the same for haploidentical HSCT and HLA-matched donor HSCT). There is almost no difference in the incidence of GVHD between haploidentical HSCT and HLA-matched donor HSCT. In accordance with the reviewer's comments, the content of the Discussion section has been supplemented and written as follows.

“The fact that the PTCy group had a high number of haploidentical HSCT patients and haplo-identical HSCT is disadvantageous in terms of the occurrence of GVHD compared to HLA-matched donor HSCT, but there were no significant differences in clinical outcomes, suggests that the use of PTCy is more effective in preventing GVHD by preserving Treg cells.”

  1. Materials and Methods

Although I understand that the journal’s formatting requirements may dictate the current structure, I would recommend considering a more conventional format, where the Materials and Methods section is presented before the Results and Discussion. This sequence helps ensure that readers first understand the experimental design and methodology before interpreting the findings and drawing conclusions, enhancing the overall readability and logical flow of the manuscript.

Author’s answer) Thank you for your feedback. We will forward your comments to the IJMS editor.

Lines 259 to 261: Is there any rationale or literature supporting the choice of the specific doses (e.g., 50 mg/kg for PTCy, 1.5 mg/kg for ATG) used in this study? It could help to briefly explain whether these doses were based on institutional protocol or evidence from previous studies.

Author’s answer) In accordance with the reviewer's comments, the source of the protocol in method section has been supplemented and written as follows.

“PTCy was given on days +3 and +4 at a dose of 50 mg/kg according to Johns Hopkins protocol. Rabbit ATG (thymoglobulin; Sanofi-Aventis, Paris, France) was given from days -3 to -1 at a dose of 1.5 mg/kg based on previous pivotal study.”

Round 2

Reviewer 1 Report

Comments and Suggestions for Authors

While the authors attempted to address my concerns, their response fails to resolve the limitations of the study raised in my first comments. I recommend rejection.